# Missed Opportunities for Antifungal Stewardship during the COVID-19 Era

**DOI:** 10.3390/antibiotics12091352

**Published:** 2023-08-23

**Authors:** Brandon K. Hawkins, Samantha D. Walker, Mahmoud A. Shorman

**Affiliations:** 1College of Pharmacy, University of Tennessee Health Science Center, Knoxville, TN 37920, USA; 2Department of Pharmacy, University of Tennessee Medical Center, Knoxville, TN 37920, USA; 3Division of Infectious Diseases, University of Tennessee Medical Center, Knoxville, TN 37920, USA

**Keywords:** antifungal stewardship, antimicrobial stewardship, aspergillosis, COVID-19, fungal infections, intensive care unit, ICU, isavuconazonium

## Abstract

Significant increases in antibacterial use were observed during the COVID-19 pandemic. However, subsequent analyses found this increase in antibiotic use to be excessive in comparison with the relatively low rates of bacterial coinfection. Although patients who are critically ill with COVID-19 may be at an increased risk for pulmonary aspergillosis, antifungal use in these populations remained underreported, particularly in later phases of the pandemic. This single-center, population-level cohort analysis compares the monthly use rates of mold-active antifungal drugs in the medical intensive care unit during April 2019–March 2020 (baseline) with those during April 2020–November 2022. The antifungal drugs included in the analysis were liposomal amphotericin B, anidulafungin, isavuconazonium, posaconazole, and voriconazole. We found that during 2020–2022, the usage of antifungal drugs was not significantly different from baseline for all included agents except isavuconazonium, which was used significantly more (*p* = 0.009). There were no changes in diagnostic modalities between the two time periods. The reported prevalence of and mortality from COVID-19-associated pulmonary aspergillosis (CAPA) may have resulted in higher rates of prescribing antifungal drugs for critically ill patients with COVID-19. Antimicrobial stewardship programs should develop and apply tools to facilitate more effective and appropriate antifungal use.

## 1. Introduction

The principles of antimicrobial stewardship experienced significant setbacks during the COVID-19 pandemic. Multiple reviews note significant increases in the prescribing of antibacterial drugs to prevent potential bacterial pneumonia in cases of COVID-19, despite reportedly low rates of bacterial coinfection (3.5–14.3%) [1]. However, qualitative research shows that high clinical suspicion of an infection is not the only factor to influence the prescribing of antibiotics by inpatient physicians. Even when a provider may not feel that antibiotics are necessary, the institutional prescribing culture and provider concern of missing an infection can spur their use [2].

Over the course of the pandemic, reports emerged of COVID-19 pneumonia being complicated by fungal superinfections [3]. Although invasive candidiasis, cryptococcosis, mucormycosis, pneumocystosis, and endemic fungal infections have been reported with COVID-19, cases involving invasive pulmonary aspergillosis (IPA) were widely disseminated throughout the global population [3,4,5]. These case reports were especially concerning for their occurrence in patients without immunocompromise, a typical hallmark of IPA [6,7]. While invasive aspergillosis has been associated with other viral illnesses, particularly in patients who are critically ill with influenza, its association with COVID-19 was relatively novel [8,9,10]. COVID-19-associated pulmonary aspergillosis (CAPA) develops via a mechanism that is not fully understood, but likely involves a complex interplay of disease, host, and iatrogenic-related factors. Some of the earliest therapies demonstrated to reduce COVID-19 mortality were corticosteroids, the use of which is normally avoided when treating influenza patients due to the potential for invasive pulmonary aspergillosis [11]. Subsequent studies also yielded reductions in the mortality of more severely ill COVID-19 patients using baricitinib or tocilizumab [12,13], both of which bear black box warnings for increased risk of fungal superinfection [14,15]. Thus, many patients were eligible to receive one or more of these agents prior to or during an ICU admission for COVID-19, potentially placing them at increased risk of developing CAPA.

Consistent with American and European guidelines for the management of aspergillosis, initial therapeutic reviews on CAPA emphasized early initiation of antifungal therapy in critically ill patients with a high suspicion for IPA [5,16,17]. These guidelines underscore the diagnostic use of histology, microbiology, clinical factors, imaging, and serologic assays (i.e., β-D-glucan, galactomannan). However, delays in receiving results for the *Aspergillus* galactomannan (GM) assay and reservations about performing aerosolizing procedures such as bronchoalveolar lavage (BAL) made it increasingly difficult to confirm suspected cases of CAPA beyond the patient’s clinical condition and radiographic findings. This diagnostic uncertainty was further complicated by poor clinical response despite best available therapy, difficulties in diagnosing mold infections, and early clinical reports of relatively high rates of invasive pulmonary aspergillosis (up to 39% in some cases) in patients critically ill with COVID-19. Consequently, concern for potential CAPA in these patients likely contributed to increased use of antifungal agents in intensive care populations.

While most reviews to date have focused on antibacterial use during the COVID-19 pandemic, we hypothesize that the pandemic also impacted the use of antifungal agents in critically ill patients. Limited data from Spain and France found elevation in the use of several mold-active antifungals, including amphotericin B, echinocandins, and voriconazole [18,19]. To date, only a single review analyzed trends of antifungal usage in the United States. These authors found lower rates of micafungin use during April to May 2020, but no significant overall difference from average intensive care unit (ICU) antifungal usage the year prior [20]. However, the short timeframe of this study lacks consideration for regional and temporal COVID-19 trends and hospitalizations. An extended timeframe review could provide additional insight into the impact of hospitalizations and the implications of CAPA on antifungal use in the later phases of the COVID-19 pandemic.

## 2. Results

Overall, we found that monthly days of therapy (DOTs) of antifungal agents per 1000 patient days (PDs) in the medical intensive care service were broadly similar between the two time periods. There was a statistically significant increase in monthly DOTs/1000 PDs of isavuconazonium (ISA) from baseline (1.05 vs. 21.21, *p* = 0.009) during 2020–2022, but there was no statistically significant difference from baseline in the usage of liposomal amphotericin B (L-AMB), anidulafungin (AFG), posaconazole (POS), or voriconazole (VRC) during this period. However, monthly usage of POS and VRC was numerically higher during 2020–2022 than during the baseline period, with DOTs per 1000 PDs being approximately 1.5× and 2× higher than baseline, respectively (Table 1). To more accurately define the usage of antifungal agents during the COVID-19 pandemic, we examined four time periods that exhibited the highest numbers of COVID-19 patient days in the medical intensive care service (July 2020, December 2020, August to October 2021, and January to February 2022). We found that only the latter two time periods (August to October 2021 and January to February 2022) had obvious increases in antifungal DOTs/1000 PDs (Figure 1). We also reviewed the results of GM testing in serum and BAL samples from patients admitted to the medical intensive care service with a diagnosis code relating to acute hypoxic respiratory failure, COVID-19, pneumonia, respiratory distress, sepsis/septic shock, and shortness of breath (Figure 2). The most common immunoassay performed for the detection of aspergillus was serum GM. Greater than 95% of serum GM assays (263/275) had an index of 0.24 or less, while approximately 87% of BAL GM specimens (59/68) had an index of 0.49 or less. Of those patients from whom a serum or BAL GM sample was obtained, only 6% (17/288) had a lower respiratory sputum or BAL specimen that was positive for presumptive *Aspergillus* species. An additional seven patients without a serum or BAL GM had a respiratory culture (sputum or BAL) positive for presumptive aspergillus. In total, 24 patients with a lower respiratory sputum or BAL culture were identified as being presumptively positive for aspergillus, regardless of whether a GM index was ordered (Figure 3).

## 3. Discussion

Our study yielded mixed results regarding the current literature reporting antifungal use in COVID-19 patients. Similar to the prior report from the U.S., antifungal usage was relatively low in the initial phases of the pandemic. [20] However, by March 2021, there had been a large increase in antifungal use among medical intensive care patients, consistent with previous reports from European studies [18,19]. In reviewing the overall trend (Figure 1) and comparing it with baseline usage, the likely reason for this surge was the increased use of isavuconazonium. The results are particularly interesting when reviewed in the context of the burden of COVID-19 on the medical critical care unit. While some timeframes more closely reflect antifungal use and COVID-19 census than others, the pandemic period was characterized by occasional decreases (August 2021 to October 2021 and January 2022 to February 2022) and increases (June 2022 to July 2022 and October 2022) in the prescribing of antifungal agents. The delayed increase in antifungal use during 2020 and into 2021 could be the result of a heightened awareness of the risk of CAPA in this patient population. This increased awareness and guidelines that advised early empiric anti-aspergillus therapy likely contributed to the increase in the use of antifungal agents, particularly isavuconazonium. This may help explain some of the increased antifungal DOTs per 1000 PDs following the admission of large waves of COVID-19 patients to the medical critical care unit, where clinical non-responders may have prompted the reevaluation of a CAPA diagnosis. The benefits of antifungal therapy in these patients may have been judged to outweigh the associated risks. While the true prevalence of CAPA remains unknown, a recent meta-analysis found that its prevalence in critically ill patients was much lower than previously reported (~10%) [21]. Inconsistent definitions of CAPA early in the pandemic may have exaggerated the reported disease prevalence and concern, leading to excessive use of antifungal agents in susceptible populations.

It is generally accepted that early reports (prior to September 2020) overestimated the rates of CAPA among critically ill patients [22]. Despite several months of high COVID-19 census in the medical intensive care unit prior to September 2020, our data suggest that the prescribing of antifungals did not begin to increase until March 2021. By this time, case reports of COVID-19 associated candidemia and mucormycosis had also begun to emerge. Based on our population level data, we cannot determine whether these additional disease states significantly contributed to the increased antifungal usage observed in the medical intensive care unit during time. However, if COVID-19-associated candidemia was a significant contributor to antifungal usage, we would expect to find an increase in the prescribing of echinocandins. These agents are preferred to azole antifungals or amphotericin B in critically ill patients for the treatment of candidemia, with limited exceptions [23]. As monthly rates of anidulafungin usage did not differ significantly from baseline during this period, we do not believe that reports of COVID-19-associated candidemia were driving the prescribing of antifungals during this period. Several of the antifungal agents included in this analysis also have varying levels of activity against Mucorales, notably liposomal amphotericin B, isavuconazonium, and posaconazole. While there was no significant change in usage of liposomal amphotericin B or posaconazole, isavuconazonium usage was significantly (~20-fold) higher than baseline. Given this, it could be inferred that the use of isavuconazonium may have been driven, at least in part, by reports of COVID-19-associated mucormycosis (CAM). Although these cases were reported globally throughout the later phases of the pandemic, the vast majority were confined to India during the summer of 2021 [3,24]. While isavuconazole usage was relatively increased in August of 2021, other instances of high monthly utilization occurred during the fall of 2021 and winter 2022, outside of the timeframe observed for most other cases of CAM. Though the exact prevalence of CAM is unknown, it appears to be less prevalent in the U.S., as compared to India. A recent review identified only 8 cases of CAM in the U.S., as compared to 42 cases from India [25]. Although we cannot conclude that reports of CAM did not impact isavuconazonium usage at our center in the U.S., we surmise that these reports presented less of a concern than those regarding CAPA.

Serologic and microbiologic testing remains at the heart of the current diagnostic methodology for invasive aspergillosis. The European Organization for Research and Treatment of Cancer/Mycoses Study Group Education and Research Consortium (EORTC/MSGERC) criteria used for the diagnosis of invasive fungal diseases utilize categories of proven, probable, and possible IPA to distinguish infection from colonization These criteria rely heavily on invasive sampling, histopathologic examination, and the testing of BAL samples for *Aspergillus* GM to confirm suspected cases of invasive aspergillosis. [26] During the COVID-19 pandemic, aerosolizing bronchoscopy procedures were avoided for the safety of the medical staff, which presented unique diagnostic challenges in confirming CAPA. While GM assays can also be performed with serum, their low sensitivity limits their usefulness in accurately ruling out the diagnosis of CAPA [27,28]. These diagnostic limitations may be further compounded by logistical considerations, as not every facility is equipped to perform on-site GM testing. At the authors’ institution, turnaround times from sample obtainment to result could take up to seven business days, further lengthening the duration of empirical antifungal therapy. Several of these factors led to a diagnostic paradox, with many patients appearing clinically ill enough to warrant empiric treatment with antifungal agents because of their risk for CAPA, but with no optimal way to accurately diagnose or rule out CAPA. This is largely reflected in the results of GM assays performed with BAL versus serum samples. For the reasons described above, serum GM indices were much more widely obtained than were BAL GM indices and the results were overwhelmingly negative for *Aspergillus* spp. (Figure 2). Interestingly, even when a BAL GM index was obtained, its performance was relatively poor compared to traditional culture techniques. It remains unknown whether some of the culture positive cases could have represented potential *Aspergillus* colonization. However, our comparison of positive respiratory cultures among all patients for whom a GM assay was obtained (Figure 3) suggests that concerns regarding the prevalence of CAPA may have been overstated. Although it is impossible to draw this conclusion definitively in the absence of individual patient level review, the authors hypothesize that many of these cases would likely fail to qualify for even the proposed “possible” category of CAPA diagnosis [27].

In the absence of high quality serologic or histopathologic evidence to support a diagnosis of CAPA, there is intense interest in developing scoring and prediction algorithms to help identify those most at risk for CAPA. However, despite the knowledge related to CAPA and other invasive fungal diseases that was accumulated during the COVID-19 pandemic, there remains a lack of standardization and consensus related to diagnostic approach, risk factors, and treatment. Numerous research definitions for CAPA have been proposed with little overlap and an overall lack of specificity [21]. Multiple international societies have published position papers regarding CAPA diagnosis and treatment, all of which acknowledge the challenge of accurately diagnosing the infection with few proven strategies for doing so [27,29]. Several studies attempted to identify specific risk factors for CAPA with results that were heterogeneous and sometimes conflicting [30]. In an effort to stratify patients at high risk for CAPA upon admission to the ICU, Calderón-Parra et al. (2022) developed and validated a local clinical prediction scoring tool they called the CAPA score at a single center tertiary institution in Madrid, Spain [31]. At their institution, age, active smoking status, chronic respiratory disease, chronic kidney disease, chronic corticosteroid treatment, tocilizumab treatment, and an elevated acute physiology and chronic health evaluation (APACHE II) score were identified as risk factors for development of CAPA [31]. Although investigators in the U.S. have proposed other diagnostic algorithms, they lack external validation [32]. While scoring algorithms and prediction tools provide important steps in delineating risks among critically ill patients, these models warrant further validation across diverse populations prior to external adoption.

Despite the high mortality associated with CAPA, it is prudent to consider that no antimicrobial agent is benign. The benefits of antifungal treatment must be judiciously weighed against the risk for potential toxicity and development of resistance. Unnecessary antifungal use puts patients, particularly those who are critically ill, at high risk for avoidable adverse events. Overuse of other, less toxic agents can impact financial costs associated with drug treatment and, in some cases, risk compromising efficacy. Several of these agents, notably the azole antifungals, have the potential for serious drug-drug interactions and toxicities associated with their use. The preferred first line treatment for invasive aspergillosis, voriconazole, is frequently associated with cardiac arrhythmias, neurotoxicity, and hepatotoxicity [33,34,35]. These adverse events may still occur despite appropriate therapeutic drug monitoring. [36,37] Significant drug-drug interactions also exist with voriconazole, making its use and management in critically ill patients complex and challenging. Other treatment options such as liposomal amphotericin B exhibit significant renal and hepatotoxicity [38]. While echinocandins are well tolerated, their use is recommended only in cases of salvage therapy or intolerance of other first line agents. Posaconazole and isavuconazonium are viable alternatives for treatment but are more expensive [39,40].

Much like with antibacterial agents, antimicrobial stewardship programs (ASPs) can play a significant role in the responsible use of antifungal drugs. The core elements of antibiotic stewardship, including prospective audits and feedback, development of locally validated screening tools and treatment protocols, and preauthorization to minimize unnecessary antibacterial use, are easily adaptable to antifungal therapy [41]. ASPs can also focus on interventions that are likely to be the most useful in determining the need for antifungal therapy, namely diagnostic stewardship and appropriate test interpretation. When supported with adequate diagnostics, ASPs improve the use of appropriate antifungal drugs and thereby reduce their economic burden [42,43,44,45,46]. Ensuring adequate testing of appropriate specimens, for example using BAL rather than serum samples whenever possible to determine GM indices, and proper interpretation of the results can help to distinguish IPA from other diseases of non-fungal concern. In retrospect, the careful review and application of risk factors associated with the development of CAPA might have reduced unnecessary exposure to antifungal agents and accompanying toxicities/expenses.

Our review of antifungal usage in ICU patients during the COVID-19 pandemic is not without limitations. Our analysis is limited to a single-center, retrospective, population level review in a limited subset of critically ill patients. This study also lacks patient level evaluations related to immunosuppressive therapy (i.e., tocilizumab or baricitinib use), length of stay, mechanical ventilation history, outcomes, drug related adverse events, and specific diagnoses. However, the diagnosis of CAPA remains difficult, in part because its clinical definitions are highly variable. Considering this and the frequent use of empiric antifungal coverage at the authors’ institution, we believe that rates of confirmed, probable, and possible IPA or CAPA are likely to add little to the discussion surrounding changes in prescribing practices and antifungal use in this patient population. Our analysis was also limited by our inability to definitively rule out the use of antifungals for other COVID-19-associated fungal infections (i.e., COVID-19-associated candidemia and mucormycosis) that might occur concomitantly with CAPA. However, given differences in locale, appropriate therapy, and associated timeframes, we believe that these were minor factors driving the usage of selected antifungals at the authors’ institution.

## 4. Materials and Methods

To investigate the potential impact of the COVID-19 pandemic on antifungal use, we examined antimicrobial usage data associated with the institution’s medical intensive care service. Days of therapy (DOT) per 1000 patient days (DOT/1000 PDs) were the units selected for evaluation, as these are currently the standard employed by the CDC’s National Healthcare Safety Network Antimicrobial Use and Resistance module. A day of therapy is defined as one calendar day (24 h) in which a specific antimicrobial is administered to an individual patient, regardless of dosing [47]. These data were reviewed for the use of five common mold-active antifungal agents, liposomal amphotericin B (L-AMB), anidulafungin (AFG), isavuconazonium (ISA), posaconazole (POS), and voriconazole (VRC). Monthly epidemiologic data was used to evaluate COVID-19-related use of antifungal agents in the ICU. As *Aspergillus* GM testing is one of the primary assays used for the diagnosis of aspergillosis at the authors’ institution, GM test data were also reviewed for patients admitted to the medical intensive care service. BAL and additional sputum specimens were also reviewed.

While other intensive care areas provided additional capacity in times of high COVID-19 caseloads, medical intensive care was the primary service involved in the treatment of critically ill COVID-19 patients at the authors’ institution. This service is exclusively staffed by the pulmonary critical care provider group, which provided a relatively consistent approach to prescribing practices. Our hospital is a 710-bed tertiary referral and academic medical center in eastern Tennessee in the U.S. that serves as one of the primary centers of acute care for the surrounding areas, including eastern Tennessee and portions of southeastern Kentucky and western North Carolina.

A two-sample T-test of unequal variances was employed to test the hypothesis that monthly antifungal usage in the medical intensive care service between April 2020 and November 2022 differed from baseline. Baseline antifungal usage for the medical intensive care service was calculated on a monthly basis using the DOT/1000 PDs from April 2019 through March 2020. This date range was selected to consider seasonality associated with prescribing trends and disease occurrence, as well as its temporal proximity to subsequent COVID-19 mitigation measures and local surges. An F-test was used to evaluate statistical variances and select the appropriate test based on the distribution. All antifungal agents exhibited unequal variances between the two time periods.

## 5. Conclusions

Isavuconazonium usage in our medical intensive care patients increased significantly from baseline following the start of the COVID-19 pandemic in the U.S. Voriconazole and posaconazole also experienced numerically higher rates of usage, but these were not statistically significant. This suggests that during the COVID-19 pandemic, the use of antifungal drugs, like that of antibacterial agents, increased as a result of various COVID-related factors. Initial reports that suggested high rates of CAPA prevalence likely exaggerated the clinical suspicion and concern for the disease. This, coupled with diagnostic uncertainty and risk-benefit analyses that favored the use of antifungal drugs in poorly responding patients, most likely led to increased antifungal use in critically ill patients.

At present, the limited available literature makes it difficult to appropriately assess the scale of antifungal use in critically ill patients with COVID-19. Further studies should be conducted to elucidate the use of antifungal agents in this patient population, particularly those with extended durations of analysis from across the U.S. Beyond antifungal use, issues remain regarding the diagnosis of IPA and CAPA. At present, definitions and diagnostic limitations complicate the accurate identification of patients at high risk for CAPA. Consensus definitions should be more widely developed to help guide appropriate treatment and determine which patients are most likely to benefit from empiric antifungal therapy. Institutions may also consider the use of alternative diagnostics given the limitations associated with *Aspergillus* GM indices. Research efforts to standardize the diagnosis and risk factors of CAPA will aid in the selection of patients likely to benefit from empiric antifungal treatment while mitigating unnecessary treatment of low-risk patients.

## Figures and Tables

**Figure 1 antibiotics-12-01352-f001:**
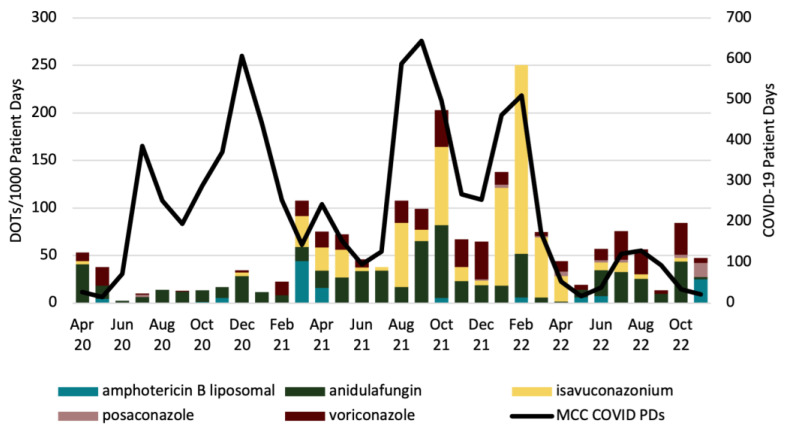
Total Monthly Medical Intensive Care Antifungal DOTs/1000 PDs for COVID-19.

**Figure 2 antibiotics-12-01352-f002:**
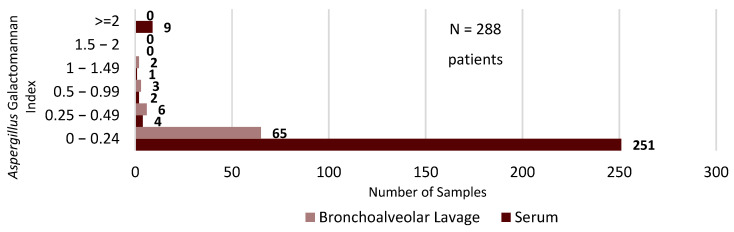
Serum vs. bronchoalveolar lavage galactomannan samples of ICU patients, 2020–2022.

**Figure 3 antibiotics-12-01352-f003:**
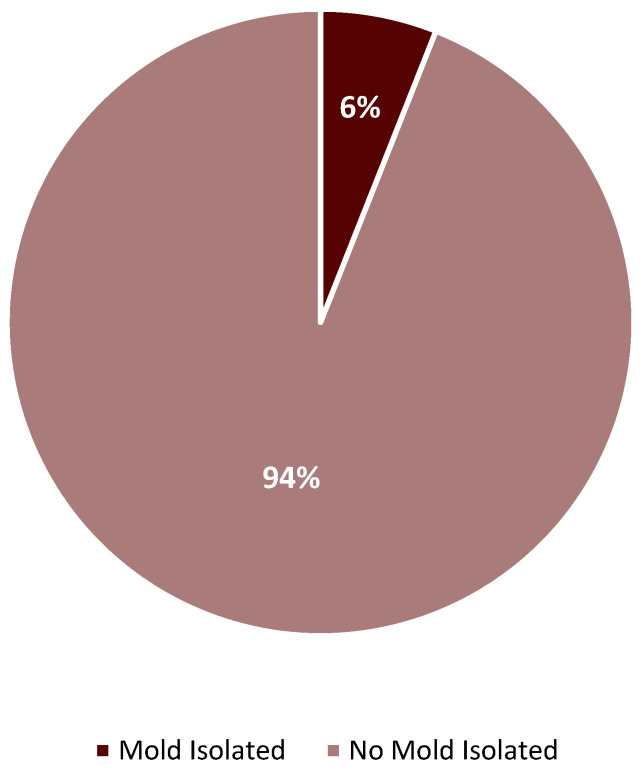
Respiratory cultures with presumptive *Aspergillus* spp. among patients with any GM obtained.

**Table 1 antibiotics-12-01352-t001:** Average Antifungal Days of Therapy per 1000 Patient Days in Medical Intensive Care.

Month	L-AMB	AFG	ISA	POS	VRC
Apr 2019–Mar 2020 Monthly Average (Baseline)	5.82	23.15	1.05	0.79	6.66
Apr 2020–Nov 2022 Monthly Average	3.70	22.12	21.87	1.17	12.71
*p*-value	0.357	0.815	0.008	0.597	0.051

Abbreviations used: AFG, Anidulafungin; ISA, Isavuconazonium; L-AMB, Liposomal Amphotericin B; POS, Posaconazole; VRC, Voriconazole.

## Data Availability

The data presented in this study are available on request from the corresponding author. The data are not publicly available due to the use of identifiable private information or identifiable biospecimens.

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
