# Peer review of "Missed Opportunities for Antifungal Stewardship during the COVID-19 Era"

_antibiotics, 2023, doi:10.3390/antibiotics12091352_

Round 1

Reviewer 1 Report (Previous Reviewer 4)

The authors addressed all my comments. The study presents some limitations, which are mainly related to its design.

I have no further comments.

Author Response

We appreciate the reviewer's previous feedback.

Reviewer 2 Report (Previous Reviewer 1)

1.    The exact categorization of enrolled patients (demographic, clinical ,radiological, comorbid status) in this study is not clearly addressed. Proper categorization and assessment of vulnerable factors is very crucial in assessment of invasive fungal infection in COVID-19 patients .

2.    Risk stratification of patients in low and high-risk group regarding development of IPA is of utmost importance as it is a guide tool for proper decision regarding antifungal therapy in critically ill covid patients .Proper risk stratification aspects not focussed in this study ( like use of low dose /high-dose corticosteroids / use of tocilizumab, baricitinib.

3.    Invasive pulmonary aspergillosis is usually diagnosed by histopathologic evidence of tissue invasion in a sterile biopsy sample, whereas in this study, only blood galactomanan is considered, which is of little significance. Sputum and BAL galactomanan or fungal culture results were not clearly discussed.

4.    The study highlighted that there was monthly increase in posaconazole usage, in COVID-19 pandemics but exact dosage and duration and indications was not mentioned. Posaconazole being a drug of choice in mucormycosis and underlying reason of its rampant use is not clearly understood, in clinical suspicion of invasive aspergillosis in critically ill covid patients .

5.    Length of stay in hospital, length of ICU stay, mechanical ventilation history, possible outcomes of enrolled patients and adverse drug reactions of antifungal agents not discussed properly, which may modulate antifungal stewardship in critically ill COVID patients.

1.    The exact categorization of enrolled patients (demographic, clinical ,radiological, comorbid status) in this study is not clearly addressed. Proper categorization and assessment of vulnerable factors is very crucial in assessment of invasive fungal infection in COVID-19 patients .

2.    Risk stratification of patients in low and high-risk group regarding development of IPA is of utmost importance as it is a guide tool for proper decision regarding antifungal therapy in critically ill covid patients .Proper risk stratification aspects not focussed in this study ( like use of low dose /high-dose corticosteroids / use of tocilizumab, baricitinib.

3.    Invasive pulmonary aspergillosis is usually diagnosed by histopathologic evidence of tissue invasion in a sterile biopsy sample, whereas in this study, only blood galactomanan is considered, which is of little significance. Sputum and BAL galactomanan or fungal culture results were not clearly discussed.

4.    The study highlighted that there was monthly increase in posaconazole usage, in COVID-19 pandemics but exact dosage and duration and indications was not mentioned. Posaconazole being a drug of choice in mucormycosis and underlying reason of its rampant use is not clearly understood, in clinical suspicion of invasive aspergillosis in critically ill covid patients .

5.    Length of stay in hospital, length of ICU stay, mechanical ventilation history, possible outcomes of enrolled patients and adverse drug reactions of antifungal agents not discussed properly, which may modulate antifungal stewardship in critically ill COVID patients.

Author Response

Reviewer 3 Report (Previous Reviewer 2)

In this revised version, the paper was submitted as an article, while in the previous version, it was a communication. In this way, more information should be included in the text. Although the authors say that the cohort involved patients from a medical intensive care unit, it is not mentioned anywhere in the text where this medical unit is located. Although it is mentioned “At the author’s institution”, there is no detailed explanation of where this institution is, and if it is a reference hospital for infectious diseases. Moreover, there is no description of the patients´ features, data, and clinical outcomes. Is there any correlation between the antifungal therapy and clinical outcome? Did it decrease the days of hospitalization?

The authors analyzed 288 patients, how many died e how many were discharged? How many used antifungals?

What about the other COVID-associated fungal infections such as candidiasis and mucormycosis? Are there any reports on those diseases in the evaluated medical unit? This should be discussed in the text.

Line 45: change “whose” by “which”.

In Figure 1, what does the symbol 月 mean?

In Figure 2, wWhat does the y axis in the graphic represent? Is it a random value for GM?

Please indicate in the results section, the figure 3 (I believe in the line 103).

In line 170, explain what is “diagnostic Catch-22”.

Round 2

Reviewer 2 Report (Previous Reviewer 1)

May be accepted 

Paper may be accepted d

Reviewer 3 Report (Previous Reviewer 2)

This revised version is suitable for publication. 

This manuscript is a resubmission of an earlier submission. The following is a list of the peer review reports and author responses from that submission.

Round 1

Reviewer 1 Report

In this study, the authors have compared the usage of five antifungal agents in intensive care units in the initial and later part of Covid-19 pandemic. The empirical use was significantly higher in the last three quarters of the year 2021 when there was greater awareness about CAPA.

While the study points out the very important subject of antifungal stewardship, it is desirable to have details of the criteria used for treatment initiation, duration of the antifungal agents and the adverse effects noted. Increased empirical use of antifungal is a well-established fact and this study does not contribute any additional information to the existing literature

Reviewer 2 Report

This communication article is a wakeup call to the use of empiric antifungal therapy in COVID-19 patients, which discuss data from one healthcare institution in Tennessee. It is a well written and discussed article, and although there are only few data, which is mentioned by the authors as a limitation of their study, it is an extremely important study, that even with limited data, it reflects the reality of the antifungal therapy and the difficulty in diagnosis of IPA and CAPA. The authors also discuss the probable consequences of the increased antifungal prescription. In table 1, please include the meaning of DOT (days of therapy) to the footnote. It is of extremely importance to report this issue in several medical and research centers.

Reviewer 3 Report

Dear Editor,

Thank you for the opportunity to review this manuscript.

The authors have highlighted the need for antifungal therapy in specific COVID-19-infected populations, as well as the challenges related to appropriate diagnosis of fungal infections and stewardship of antifungals - in the context of the COVID-19 pandemic. This is a very important subject given the resurgence of COVID-19 infection in various parts of the world, and the need for appropriate stewardship of antimicrobials for improved treatment and outcomes. I commend the authors for undertaking this very important study, and for highlighting gaps that need to be addressed.

The authors compared use of specific antifungal agents at a tertiary referral and academic centre – before and during the COVID-19 pandemic. The extended timeframe employed to explore for changes in antifungal use during the pandemic enabled evaluation of antifungal usage over time – highlighting increased use further on in the pandemic. The authors utilised a standardised reporting metric, days of therapy per 1000 patient days (DOT/1000 patient days), for comparison of antifungal use before and during the pandemic.

I have one question, please: was there a possibility of comparing the patient populations who received antifungal therapy? If so, was there any difference in baseline characteristics between the population who received antifungals before and during the COVID-19 pandemic  (with the exception of COVID-19 infection in the latter)?

Reviewer 4 Report

Hawkins and Walker present a study addressing changes in the prescription of anti-fungal therapy (5 different antifungals) during the COVID-19 pandemic. The study was performed in two different ICUs from a single tertiary center.

Overall, they found a substantial increase in the use of these drugs during the last 3 trimesters of 2021. The authors conclude that there were probably no substantial reasons for this variation and that an anti-fungal stewardship program could impact the use of these drugs.

I think that this study, as it is, has substantial limitations:

1- The authors correctly pointed out that pulmonary aspergillosis often complicates the COVID-19 patient, especially those submitted to long periods of invasive mechanical ventilation. However, no information was given regarding COVID-19 patients admitted to the ICU, time on invasive mechanical ventilation, and the number of patients receiving it for long periods.

2- Moreover, no information regarding the number of patients diagnosed with CAPA is provided. Patients with CAPA often received antifungals for long periods of time and that could have influenced antifungals DOT in a disproportional mode (this is quite different from antibacterial DOT).

3- Since this is a single-center study, information on diagnostic methods available for the diagnosis of CAPA should be provided, especially if changes in methods available occurred during this period.

4- Also information regarding the number of patients with positive cultures for Aspergillus should also be provided. Besides, the number of patients diagnosed with confirmed CAPA or those who actually died of CAPA should be provided, as these could have influenced prescription practices.

5- The authors stated that the fact that this is a single-center study was mitigated by the fact that this is a referral center. However, the study is evaluating prescription practices and not diseases. These practices are only related to the center and not to the type of patients. This should be clearly stated in Limitations.

Minor comments

1- The abstract should refer to this as a single-center study; it should also mention if there was any difference in diagnostic methods available or if there were changes in protocols for CAPA.

2- In the Discussion authors referred to the possible toxicity of antifungals. Do the authors have any data on actual adverse events in this population? 

3- Conclusions should be short sentences related to the study's main goals. Authors should refrain from using conclusions to discuss this or other studies. This belongs in the Discussion.